# Antioxidant and Antiproliferation Activities of Lemon Verbena (*Aloysia citrodora*): An In Vitro and In Vivo Study

**DOI:** 10.3390/plants11060785

**Published:** 2022-03-16

**Authors:** Hasan M. Rashid, Asma Ismail Mahmod, Fatma U. Afifi, Wamidh H. Talib

**Affiliations:** 1Department of Clinical Pharmacy and Therapeutics, Applied Science Private University, Amman 11931, Jordan; hasanalzubaidy08@gmail.com (H.M.R.); asmamahmod1212@gmail.com (A.I.M.); 2Department of Pharmaceutical Chemistry and Pharmacognosy, Applied Science Private University, Amman 11931, Jordan; fatma_afify@asu.edu.jo or

**Keywords:** lemon verbena, plant extracts, traditional medicine, antioxidant, antiproliferation

## Abstract

*Aloysia citrodora* (Verbenaceae) is traditionally used to treat various diseases, including bronchitis, insomnia, anxiety, digestive, and heart problems. In this study, this plant’s antioxidant and anti-proliferation effects were evaluated. In addition to volatiles extraction, different solvent extracts were prepared. The GC-MS, LC-MS analysis and the Foline-Ciocalteu (F-C) method were used to investigate the phytochemical components of the plant. MTT assay was used to measure the antiproliferative ability for each extract. Antioxidant activity was determined using the 2,2-diphenylpicrylhydrazyl (DPPH) assay. In in vivo anti-proliferation experiments, Balb/C mice were inoculated with tumor cells and IP-injected with ethyl acetate extract of *A. citrodora*. After treatment, a significant reduction in tumor size (57.97%) and undetected tumors (44.44%) were obtained in treated mice, demonstrating the antiproliferative efficacy of the ethyl acetate extract. Besides, ethanol extract revealed the most potent radical scavenging effect. The findings of this study displayed that *A. citrodora* has promising cytotoxic and antioxidant activities. Still, further testing is required to investigate the extract’s chemical composition to understand its mechanisms of action.

## 1. Introduction

Worldwide estimation of cancer incidence and mortality, created by the International Agency for Research on Cancer, revealed that 19.3 million new cancer cases and almost 10.0 million cancer deaths occurred in 2020. Female breast cancer is the most diagnosed cancer, surpassing lung cancer, with (11.7%) new cases of the total reported cases, followed by lung cancer (11.4%). The global cancer incidences are expected to be 28.4 million cases in 2040, a 47% rise from 2020 [1].

Cancer is a disorder that triggers normal cells’ uncontrolled growth and alters the genome (which causes malignant characteristics in normal cells) [2]. This overgrowth progression impairs the normal biological process of healthy cells by the invasion of nearby tissues and metastasizes to distant tissues [3]. The first options of treatment are chemotherapy, radiation therapy, and surgery. Many adjuvant therapies and strategies such as quality of life changes, antioxidant supplements, herbal medicines, and remedies based on natural products are continuously growing as promising strategies to augment conventional therapies [4,5,6]. Today, the most reliable option for cancer treatment is chemotherapy, and drug resistance appears as a significant limitation [7,8]. These limitations push researchers to find other sources for treatment and direct our vision into the richest source of treatments from ancient times, to nature.

Another problem is the free radicals that can cause oxidative damage to any sort of cell component. This type of damage in humans can lead to various degenerative disorders, including cancer and cell aging, directing the interest to the antioxidants [9]. Antioxidant molecules, pharmacologically effective and with few or no adverse effects, are currently being explored for preventive care worldwide and in the food industry. Plants are vulnerable to the harmful effects of active oxygen and produce many antioxidant chemicals in addition to tocopherols. Flavonoids, several other phenolic compounds, and polyphenolics are examples of these substances [9].

All societies around the world have consumed medicinal plants. In the United States and Europe, natural products or their derivatives account for nearly (50%) of all prescription drugs [10]. Many drugs used to treat cancer, infections, and other disorders are entirely derived from plants or are synthetic/semi-synthetic derivatives of plants [11]. According to the World Health Organization (WHO), (11%) of drugs are derived entirely from plants, with a significant percentage of synthetic medications derived from natural precursors [12]. Showing the increasing importance of natural products in drug development.

Approximately 150 plant herbs are still used as a source of traditional herbal therapy in Jordan [13]. One of these herbs is Lemon verbena, scientifically known as *Aloysia citrodora* Paláu, an acknowledged medicinal plant [14]. *A. citrodora* has a broad range of medicinal, cosmetic, aromatic, and culinary applications [15]. *A. citrodora* extracts and preparations are mostly reported to have antioxidant and anti-microbial biological activities [16,17,18,19]. Phytochemically, the extracts of *A. citrodora* were found to have an excessive amount of phenolics such as phenylpropanoids, flavonoids, lignans, tannins, and a variety of other nonphenolic compounds [20]. Generally, the major compounds of the oil of *A. citrodora* were the citral isomers geranial and neral [21,22]. However, limonene, 1,8-cineol, β-caryophyllene, citronellal, and citronellol were frequently listed as major ingredients. Furthermore, studies showed that the oil composition is affected by numerous factors like plants’ genotype, environmental factors, and growth conditions [23,24].

To the best of our knowledge, this study is the first to evaluate Jordanian origin *A. citrodora* regarding its LC-MS, SPME, total phenol, and cytotoxic activity in vitro and in vivo. Also, this study is the first to use ethyl acetate extract of *A. citrodora* and identify its active components. In the present study, different extracts of *A. citrodora* aerial parts were prepared and tested for their antioxidant and antiproliferation activity, in addition to various qualitative and quantitative phytochemical experiments.

## 2. Results

### 2.1. Phytochemical Studies

#### 2.1.1. Plant Samples Preparation and Percentage Yields

The results showed variations in amounts and percentage yields upon extracting 100 g of *A. citrodora* aerial parts using different solvents. The highest amount and percentage yield were obtained from water extract (5.5%), followed by ethanol (5.25%) and ethyl acetate extracts (2.3%). On the other hand, the essential oil (EO) extraction percentage yield by a hydrodistillation (HD) technique resulted in a low amount (0.25%) (Table 1).

#### 2.1.2. Determination of Total Phenolic Content (TPC)

Ethanol extract revealed the highest gallic acid equivalent amount (GAE) with a value of 117.13 mg/g, followed by ethyl acetate extract with 73.44 mg/g. in comparison, water extract revealed the lowest phenolic amount (GAE) with a value of 66.6 mg/g (Figure 1). Standard curve found in (Appendix A).

#### 2.1.3. Liquid Chromatography-Mass Spectrometry (LC-MS)

Qualitative analysis of ethyl acetate extract using LC-MS revealed the presence of six compounds. All the detected compounds were flavonoids. Naringenin was the compound with the highest concentration (25.22%), followed by 5,6,4′-Trihydroxy-7,3′-Dimethoxyflavone (23.67%), Hispidulin (22.61%), Eupatilin (13.24%), Baicalein (7.84%), and 5,7-Dihydroxy-2′-Methoxyflavone (4.91%). The identified compounds represent (97.52%) of the total ethyl acetate extract components (Table 2). The percentages are represented the comparable values of the compounds determined in the extract shown in Figure 2. (Detailed chromatogram found in Appendix A).

#### 2.1.4. Gas Chromatography-Mass Spectrometry (GC-MS) Analysis of the Hydrodistilled Oil and Aroma Profile Determination Obtained by Solid Phase Micro Extraction (SPME)

In the hydrodistilled oil, monoterpenes accounted for 60.48% of the total identified compounds (97.86%). Oxygenated monoterpenes yielded a total of (36.82%) versus hydrocarbon monoterpenes (23.66%), while the hydrocarbon monoterpene d,l-Limonene was identified as the major component (18.80%). The second major component of the hydrodistilled oil was the sesquiterpene hydrocarbon γ-Muurolene (14.13%). Still, trans-chrysanthenyl acetate (10.27%), verbenyl acetate (9.10%), 1,8-cineol (8.14%), β-caryophyllene (5.09%) and spathulenol (3.75%) occurred in remarkable concentrations.

In contrast, although the aroma profile obtained by SPME was dominated by sesquiterpene hydrocarbons (53.57%), the monoterpene hydrocarbon d,l-Limonene was identified as the major component with (34.40%). No oxygenated sesquiterpenes were detected, while the oxygenated monoterpenes occurred in a negligible amount of (0.64%). The predominant sesquiterpene hydrocarbons were γ-muurolene (17.25%) and β-caryophyllene (16.80%).

Total identified components of the fresh aerial parts of *A. citrodora* (43 compounds) accounted for (97.86%) using hydrodistillation, while the aroma profile revealed the occurrence of 27 compounds accounting for (99.81%) using the SPME method. Data are shown in Table 3. GC-MS analysis for both methods is found in (Appendix A).

### 2.2. Anti-Oxidant Activity

Different extracts of *A. citrodora* from fresh aerial parts exhibited radical scavenging activity against DPPH radical. Compared with ascorbic acid (IC_50_: 2.783 mg/mL), ethanol extract had the most potent antioxidant activity (IC_50_: 22.85 mg/mL), followed by the aqueous and ethyl acetate extracts with IC_50_’s of 49.9 mg/mL and 107.04 mg/mL, respectively (*p*-value = 0.148). On the other hand, essential oil did not exhibit any antioxidant activity using DPPH (Table 4).

### 2.3. Antiproliferative Activity

#### 2.3.1. In Vitro Studies

In vitro antiproliferative activity of different extracts of *A. citrodora* was tested on five cell lines (EMT-6, MCF7, MDA-MB231, T47D, and Vero), and the obtained IC_50_ (µg/mL) results showed promising anti-proliferation potential (Table 5). By far, ethyl acetate extract was the most active on all cell lines, followed by the essential oil and ethanol extract. On the other hand, the aqueous extract did not exhibit activity on all the tested cell lines (Figure 3).

The *A. citrodora* extracts and essential oil showed a lower toxicity effect against Vero cell line compared to the cancer cells (Figure 4).

The estimation of the selectivity index depends on the IC_50_ ratio of the normal cell line (Vero cells) versus the tumor cells, which demonstrates the selectivity of the sample to the tested cell lines. The samples are classified high selective if their SI value is more than three [25] (Table 6).

#### 2.3.2. In Vivo Experiments

##### Establishing Doses for the In-Vivo Experiment (Determination of LD_50_)

A limit study was conducted on a small group of mice to select the dose ranges for actual LD_50_ (median lethal dose). According to the Karber method, the LD_50_ of *A. citrodora* ethyl acetate extract was 1.62 g/kg (Table 7).
$${\mathrm{LD}}_{50}=\mathrm{the}\;\mathrm{least}\;\mathrm{lethal}\;-\;\sum (a*b)/\mathrm{number}\;\mathrm{of}\;\mathrm{mice}$$

##### Antitumor Effects of *A. citrodora* on EMT6/P Cells Implanted in Mice

Ethyl acetate extract, which exhibited the best results in in vitro experiments, was selected for in vivo study. A significant reduction in tumor size (*p* = 0.003) was observed in the treatment group compared to the negative control, where tumor growth was increased by (49.94%). The treatment group caused a high percentage in tumor size reduction (57.97%) and the undetected tumors (44.44%) with no recorded deaths (Table 8 and Figure 5). Comparison of tumor sizes after dissection at day 10 in all groups (*n* = 9) are shown in Figure 6.

### 2.4. Evaluation of Liver and Kidney Functions in Treated Mice

The treated group exhibited insignificant (*p* > 0.453) differences in serum ALT and (*p* > 0.92) AST levels compared to normal untreated mice. The ALT level for untreated mice that bore tumors was elevated with a value of 37.21 IU/L. In comparison, the treatment group had a value of 21.66 IU/L. On the other hand, the AST level of untreated mice bearing tumors and the treatment group was 101.65 IU/L and 98.32 IU/L, respectively. Levels of serum ALT and AST were within the normal range for the treatment group compared to the levels recorded for normal untreated mice with a value of 28.88 IU/L and 119.98 IU/L, respectively. Tumor-bearing mice treated with *A. citrodora* 0.162g/kg showed elevated serum creatinine levels (207.23 µmol/L) compared to normal mice (97.24 µmol/L). On the other hand, serum creatinine levels were in the control group that bore tumors with a value of (106.20 µmol/L) compared to normal mice. Nevertheless, the difference in normal and untreated serum creatinine levels compared to the treated group was insignificant (*p* > 0.532). (Table 9).

## 3. Discussion

*Aloysia citrodora* is a common aromatic plant that is cultivated in Jordan and many nearby countries, particularly for food and medicinal purposes. Several types of phytochemicals have been found in different parts of *A. citrodora*. Various flavonoids are isolated from methanol, ethanol, or aqueous extracts of the plant. Moreover, the essential oil is mainly composed of hydrocarbon and oxygenated mono- and sesquiterpenes [26]. Limited data are available exploring the Jordanian *A. citrodora* biologically. The current study aimed to evaluate the antioxidant and antiproliferative activities of fresh aerial parts of *A. citrodora*, growing in Jordan.

### 3.1. Antioxidant Activity

Natural antioxidants found in medicinal herbs are responsible for reducing or inhibiting the harmful effects of oxidative stress and reactive oxygen species (ROS), which have been detected in almost all cancer types and promote their progression, development, survival, and boost its metastatic ability. Novel therapeutic approaches are needed to control the intracellular ROS signaling production and ROS-induced tumor. Then again, the antioxidants could avert initiation events of cancer, where ROS considered a crucial part [27]. The hydroxyl groups of the active compounds permit a scavenging effect on various reactive oxygen and nitrogen species and provide a powerful antioxidant effect [28].

Free radical scavenging activity is determined through several available procedures. However, the radical-scavenging DPPH test has attracted the most attention [29]. It has been utilized for evaluating the antiradical properties of extracts very often because it can accommodate many samples in a short amount of time and is sensitive enough to detect active components at low concentrations [30].

Ethanol extracts were found to be the most potent scavengers against the DPPH radical, followed by water extract. It could be due to the amount of phenols and flavonoids [31]. Depending on the TPC experiment, phenols are abundant in these extracts. Although TPC measures all phenolics, some heterogeneous phenolic compounds could respond differently. F–C values are found to be higher than those obtained by other methods such as HPLC-MS/MS and HPLC-UV [32,33]. A study implied that the antioxidant capability of an extract is determined not only by the amount of polyphenols present but also by the nature of the antioxidants molecules [34]. The potent antioxidant extracts are rich in polyphenols. Numerous investigations have found a link between phenolic composites and radical-scavenging capacity [35,36]. A significant linear correlation was found between the content of phenolic compounds and the extracts’ antioxidant activity in a few plants [37].

Ethyl acetate extract showed lower activity against DPPH. The previous findings on the extract components revealed potent activity when assessed solely [38,39,40]. This activity could be affected by many factors. The constituent of the reacting mixture differs in molecular size, polarity, solubility, chemical structure, concentration, and molecular ratio. The interactions between the components (not only with the active compounds) result in additive, synergistic or antagonistic effects [41]. As in the case of ethyl acetate extract, most probably an antagonistic effect between its components causes the drop in the activity.

The free-radical scavenging activity of *A. citrodora* EO was also assessed in this study. The oil did not exhibit any radical scavenging activity at all concentrations examined. Abuhamdah, R., & Mohammed (2014) conducted a study on Jordanian origin *A. citrodora* fresh and dried leaves identified 83 compounds of essential oil chemical composition. The highest components were limonene (12.14%) caryophyllene oxide (10.44 %), curcumene (9.17 %), spathulenol (7.16 %), 1,8-cineole (7.94%), followed by geranial (4.03 %) and neral (2.55) [42]. Another study of the EOs obtained from dried or fresh Jordanian *A. citrodora* leaves identified limonene, neral, and geranial as major components. They have also identified in low amounts α-pinene, α-terpinene, sabinene, linalool, and caryophyllene [43]. A study of Jordanian *A. citrodora* aerial parts by Hudaib et al. (2013) revealed limonene (17.7%) as the highest constituent, together with the citral isomers (neral: 9.8% and geranial: 10.1%), representing more than one-third of the oil. Other major components identified in *A. citrodora* oil included mainly 1,8-cineole (11.7%), α-curcumene (6.3%), and the oxygenated sesquiterpene components spathulenol (4.6%) and caryophyllene oxide (3.1%) [44].

Studies from several countries in the literature revealed the presence of citral isomers (neral and geranial) in *A. citrodora* essential oil [45,46,47,48,49]. However, the present study results did not indicate the occurrence of any common major detected compounds (e.g., neral, geranial, thujone, citronellal, carvone). In a recent article of the *A. citrodora* EOs from two separate localities, the antioxidant activity of the plant containing more citral resulted in more potent antioxidant activities [50]. The literature emphasizes that various factors could affect EOs composition, such as the age-related stage of the plant, its physiology, and growth conditions. Also, the constituents of the EO could be altered by the isolation and analysis conditions [43,51,52,53].

Another factor affecting EOs antioxidant activity could be the presence of Alpha-pinene and limonene compounds which have poor activity in the DPPH test system. The antioxidative effect of EOs is frequently not attributable to the primary constituents; lesser molecules and synergistic effects may play a substantial role in the activity [54]. Antioxidant activity can be induced by the presence of heteroatom-containing compounds in EOs. Oxygen-containing moieties, such as phenols or hydroxyl, are more effective antioxidants than nitrogen-containing structures like aniline [55].

Variations in the amounts of each compound identified in these extracts had a strong correlation with the extract activity. Jordanian *A. citrodora* in this analysis revealed high variation in its composition compared to studies from different countries [54,55], even its EO components identified in Jordan 9 years ago [44]. The current findings could indicate the significant effects of these factors (mentioned above) on the quality and quantity of the extracts.

### 3.2. Antiproliferative Activity

#### 3.2.1. In Vitro Study

Ethyl acetate extract established the highest anti-proliferation activity, followed by the EO and ethanol extract. In contrast, water extract exhibited the lowest activity. The difference in activity could result from the various flavonoids and phenolic substances’ chemical structure types found in the extracts. Like in many other plant species, flavonoids might be responsible for the major bioactivities of A. citrodora, such as antimicrobial, neuropsychological, antioxidant, cytoprotective, and anti-cancer effects [26]. Flavonoids can occur naturally either as a compound associated with sugar in conjugated form (glycosides) or without attached sugar as aglycones. The presence or absence of sugar moiety can affect the flavonoids’ solubility, reactivity, and stability [56]. Overall, pharmacokinetic properties can have a major impact on the health-promoting effects of phytochemicals [28].

To the best of our knowledge, the current study is the first attempt to assess the antiproliferative property of the aerial parts of *A. citrodora* extracted by ethyl acetate (EA) as solvents that had the ability to extract various non-polar active compounds. The LC-MS analysis revealed naringenin (flavanones) as the most abundant compound in EA extract (25.22%). Naringenin is insoluble in water and soluble in organic solvents, like alcohol [57,58]. It has been found in several plants possessing various biological activities like antioxidant, antitumor, antiviral, antibacterial, anti-inflammatory, and cardioprotective effects [38,59,60,61]. In many reports, Naringenin exhibited antiproliferation effect, ability to inhibit cell growth, increase AMP-activated protein kinase phosphorylation, CyclinD1 expression down-regulation, and cell death induction. Other reports include promising results for prostate cancer, melanoma, and gliomas-brain cancer [62,63,64].

Another major detected component was 5,6,4′-Trihydroxy-7,3′-Dimethoxyflavone (5-TDMF) (23.67%)**.** It is proven to have potent antioxidant activity in vitro and ex vivo. One study demonstrates that 5-TDMF has potent antioxidant and anti-inflammatory effects without cytotoxicity observed. The same study suggested that (5-TDMF) could block LPS-induced NF-κB translocation and iNOS and COX-2 expressions by inhibiting the mitogen-activated protein (MAP) kinase and MAPK/ERK signaling pathways. Suggesting potential novel chemo-preventive anti-inflammatory agents [40]. Still, a lack of detailed studies is noticed regarding its molecular mechanism that controls its activities, such as binding to a particular protein structure [65].

Hispidulin (HIS) (22.61%) is another detected compound considered a monomethoxy and trihydroxy flavone compound. In vitro investigations suggested the capability to affect the activation of JNK, p38, and NF-κB [66]. Also, it could control helenalin-induced cytotoxicity [67]. HIS has been reported in numerous studies to have potential antimutagenic, antioxidative, and anti-inflammatory effects [68,69,70]. Besides, many reported anticancer activity against multiple cancer cell lines such as gastric, pancreatic, ovarian, gallbladder, and colorectal. In addition to glioblastoma renal carcinoma, acute myeloid leukemia, Glioblastoma Multiforme, hepatocellular carcinoma cancers [70,71,72,73,74,75,76,77,78,79,80,81,82,83]. In colon cancer, HIS was noticed to inhibit the hypoxia-generated epithelial-mesenchymal transition, which had significantly enhanced the cytotoxicity of anticancer drugs against cancer cells [71].

Other detected components include Eupatilin (5,7-dihydroxy-3′,4′,6-trimethoxyflavone) (13.24%). That is known to possess promising antiproliferative, anti-inflammatory, modest antioxidant, neuroprotective, anti-allergic, and cardioprotective activities [39,84,85,86,87,88,89,90,91,92]. Several studies on eupatilin have explained its anti-cancer property due to its promising capacity to prompt apoptosis in different cancer cell lines [84,85,93,94,95,96,97,98,99]. Same as baicalein bioflavonoid (S7.84%), which can arrest cancerous cells at the G2/M and G1/S cell cycle phases [57]. It also decreases cell proliferation or induces apoptosis in multiple myeloma and cancer types. It has been demonstrated to inhibit cancer cell migration and invasion in many studies [100,101,102,103,104,105].

No available studies have been found related to 5,7-Dihydroxy-2′-Methoxyflavone (2′-Methoxychrysin) (4.91%) activity, although this compound has been separated previously from many species [106,107]. This compound has the physicochemical and pharmacological effects of flavonoids—flavones, and flavon-3-ols [108]. (All the structures of the detected compounds are found in Appendix A)

Previous reports revealed different types of flavonoids identified in *A. citrodora* as Skaltsa used column chromatography to isolate several flavonoids from the leaf extract of *A. citrodora* in 1988. Flavone structures were found in all of the purified substances [109]. Subsequently, glycosides of previously isolated flavones were also detected, such as apigenin-7-diglucuronide and chrysoeriol7-diglucuronide in the aqueous extract of *A. citrodora* aerial parts [110]. New flavonoids in the aerial parts (jaceosidin, nepetin, and nepitrin) have also been reported in recent studies, all of which have flavone structures [111].

The occurrence of these compounds simultaneously could indicate a synergistic, additive, or antagonistic effect on the antiproliferative activity of the extract. Ethyl acetate extract had the most potent antiproliferative effect on all used cancer cell lines (IC_50_ ranging from 136 to 203 μg/mL), even on the normal cell line, which in comparison with cancerous cells and depending on selectivity index, revealed selective toxicity (targeting malignant cells without harmful effect on normal cells) using suitable concentration.

The antiproliferative experiments of the EO of the aerial parts of *A. citrodora* revealed weak activity compared with other studies (IC_50_ ranging from 402 to 633 μg/mL). Oukerrou et al. (2017) demonstrated that *A. citrodora* EO exerted a dose-dependent cytotoxic effect on P815, MCF7, and Vero tumor cell lines, with IC_50_s ranging from 6.60 to 79.63 μg/mL [112]. Another study observed the potent cytotoxic effect of *A. citrodora* EO from two different regions of Palestine on HeLa cell lines. The IC_50_ values were 84.50 and 33.31 μg/mL and compared with Doxorubicin (IC_50_ value of 22.01 μg/mL) [50]. No notable activity was observed regarding ethanol, and aqueous extracts in this study suggested the presence of weakly active polar compounds or antagonistic interaction between the phenolics and flavonoids components.

#### 3.2.2. In Vivo Study

The ethyl acetate extract of *A. citrodora* was used in the current study to treat mice implanted with breast cancer cells. A significant reduction in tumor size and a high cure percentage were observed. *A. citrodora* ethyl acetate extract dosage of 0.162 g/kg showed a high reduction in tumor size by 57.97% and many undetected tumors or curing effects of 44.44%. Reduction in the tumor size may be explained by the cytotoxic phytochemicals that exhibited notable effects on tumor cells.

Many studies showed the effect of phytochemical compounds detected in *A. citrodora* with cytoprotective, antioxidant, and anti-proliferation activities [26]. Naringenin, 5,6,4′-Trihydroxy-7,3′-Dimethoxyflavone (5- TDMF), and hispidulin were the most concentrated compounds in ethyl acetate extract. These compounds possess promising antiproliferative activities. In vivo studies with naringenin revealed anti-proliferation activity through a delay of tumor growth on ovariectomized C57BL/6 mice injected with E0771 mammary tumor cells [60]. In another in vivo study using breast cancer cells, observations suggested decreased secretion of TGF-β1 and accumulation of intracellular TGF-β1, and inhibition of TGF-β1 transport from the trans-Golgi network and PKC activity [113]. Also, 5,6,4′-Trihydroxy-7,3′-Dimethoxyflavone (5- TDMF), hispidulin in-vivo anti-proliferation activity, and antioxidant properties were reported [40,114]. These compounds may work synergistically to inhibit cancer either by a direct effect on cancer cells or by an indirect effect through antioxidant activity and other mechanisms [115].

Liver and kidney function enzymes such as ALT, AST, and creatinine are the most reflective parameters of toxicity and safety profile at the therapeutic doses because they are significant in eliminating the drugs through metabolism and excretion [116].

The ALT and AST for the treated group were within the normal range and lower than the normal group, which indicates an acceptable safety profile for all the treatments. These results could be justified because the doses used in this study chose a dose according to LD_50_ estimation and used Karber method calculation with no toxic outcomes. On the other hand, creatinine levels observed in this study were higher for the treatment group than the control and normal mice creatinine levels. High kidney enzymes are acceptable as long as they are within the range of normal mice enzymes level. Mainly after no deaths were observed after ten days of treatment, the results might indicate the safety of using the ethyl acetate extract.

## 4. Materials and Methods

### 4.1. Plant Material

Fresh plants of *A. citrodora* were purchased from a local plant nursery (Shafa Badran, Amman, Jordan) in July 2021. It was five meters in height, in the maturation phase with tiny lilac or white blooms that appeared during July to August flowering season [117,118,119]. Prof. Fatma Afifi (The Faculty of Pharmacy, Applied Science University) authenticated the plant using descriptive references and comparing it with the herbarium specimen hinge on the voucher specimen at the University of Jordan (herbarium number AC-V1) [118,120].

### 4.2. Plant Extracts Preparation

For this study, three different extracts were prepared: distilled water, 70% ethanol, ethyl acetate, and essential oil. Each 100 g of chopped fresh aerial parts was extracted with (1:10 *w*/*v*) distilled water, 70% ethanol, and ethyl acetate by applying gentle heating until boiling. Then the extracts were kept overnight at room temperature (RT) and filtrated. Solvents were evaporated using a rotary evaporator at 40 °C until a syrupy extract was obtained.

The essential oil was extracted by hydrodistillation technique using Clevenger-type apparatus. 300 g of aerial part of the plant was cut into small pieces and chopped using a mixer for one minute, then hydrodistilled for three hours. The oil was then collected, separated, and dried from the aqueous layer by adding anhydrous sodium sulfate (Na_2_SO_4_) and stored at 4 °C in amber glass vials until analysis [121,122].

### 4.3. Determination of Total Phenolic Content (TPC)

TPC of water, ethanol, and ethyl acetate extracts of *A. citrodora* was executed according to the (F-C) reagent procedure explained by Mahindrakar & Rathod (2020), with slight modifications [123]. Briefly, 200 µL of 1 mg/mL of each extract was diluted with distilled H_2_O (2 mL) in a 5 mL volumetric flask. Then add (F-C) reagent (200 µL) to each mixture and mix properly. After 4 min, adding 800 µL of 7.5 % Sodium carbonate (Na_2_CO_3_) followed by volume adjusting to 5 mL with DH2O. A dark blue color was formed, and the obtained mixture was incubated for 1 h in a dark place at RT. A spectrophotometer measured the absorbance of all samples at 765 nm. Gallic acid (GA) gave the calibration curve standard, using five serial dilutions (100, 50, 25, 12.5, and 6.125 μg/mL). Each reading was done in triplicate, and methanol was the blank. Total phenol content was valued as gallic acid equivalent in mg for each gm of the extract (GAE).

### 4.4. Liquid Chromatography-Mass Spectrometry (LC-MS)

Ethyl acetate extract as a sample has been dissolved with 2.0 mL DMSO then completed to 50 mL by acetonitrile and centrifuged each sample at 4000 rpm for 2.0 min. Then 1.0 mL was transferred to autosampler, and 3.0 µL was injected (All standards were used to identify ms/z and the retention time). The testing was executed using Burker Daltonik (Bremen, Germany) impact II ESI-Q-TOF system set up with Burker Dalotonik Elute UPLC system (Bremen, Germany). The instrument used the Ion Source Apollo II ion funnel electrospray supplier (capillary voltage: 2500 v; nebulizer gas: 2 bar; dry gas flow: 8 L/min; dry temperature: 200 °C; mass accuracy: below 1ppm; mass resolution: 50,000 FRS; the TOF repetition rate: 20 kHz). Chromatographic separation was achieved via Burker solo 2-C-18 UHPLC column (100 mm × 2.1 mm × 2 µm) at a flow rate of 0.51 mL/min and a column temperature of 40 °C.

### 4.5. Gas Chromatography-Mass Spectrometry (GC-MS) Analysis

Chemical composition examination of the volatile mixtures was performed through diluting about one μL aliquot of each oil sample to 10 μL in GC grade hexane and exposed to a GC/MS analysis utilizing a Varian Chrompack CP-3800 GC/MS/MS-200 (Saturn, Amsterdam, The Netherlands), coupled with a DP-5 (5% diphenyl, 95% dimethyl polysiloxane) GC capillary column (30 m × 0.25 mm i.d., 0.25 μm film thicknesses) and helium (the carrier gas) with a flow rate of 0.9 mL/min. The MS source temperature and the ionization voltage were regulated to be 180 °C and 70 eV, respectively. The starting temperature of the column was 60 °C for 1 min (isothermal), followed by gradual temperature rising (3 °C/min) to 250 °C. The recognition of the main compounds is achieved either by matching to built-in libraries (NIST Co. and Wiley Co., MD, USA) or comparing their retention index (RI) to C8-C20 n-alkanes literature values analyzed with columns of identical polarity or authentic samples (thujene, sabinene, α-and β-pinenes, myrcene, limonene, p-cymene, γ-terpinene, terpinolene, trans-caryophyllene, and germacrene D [Sigma-Aldrich, St. Louis, MO, USA]). The quantitative analysis was conducted via Hewlett- Packard HP-8590 gas chromatography coupled with a split-splitless injector (split ratio 1:50) and a Flame ionization Detector (FID). The column was an optima-5 (5% diphenyl, 95% dimethyl polysiloxane) fused silica gel capillary column (30 m × 0.25 mm, 0.25 μm film thickness). Regulation of the oven temperature was set as a rate of 10 °C/min from 60 °C to 250 °C and then kept stable at 250 °C for 5 min. The estimation concentration of the identified compounds was established relying on the calculated relative peak areas of the oil components.

### 4.6. Solid Phase Micro Extraction (SPME) Method

Supelco SPME devices coated with (poly-dimethylsiloxane) (PDMS, 100 µm) were used. The fiber was exposed to the fresh plant samples in tightly closed amber glass vials for three minutes. Once sampling was finished, the fiber has retained into the needle and moved to the injection port of the GC and GC-MS system, operated in conditions identical to the hydro-distilled oil for both quantification and detection of the constituents. The compounds were identified using the built-in libraries data by comparing mass spectra and retention index with those in literature as mentioned for HD.

### 4.7. Antioxidant Activity

The 2,2-diphenyl-1-picrylhydrazyl (DPPH) radical was used to test the radical-scavenging activity of *A. citrodora* extracts and essential oil following the method of Brand-Williams [124]. Briefly, different concentrations (400–0.19 µg/mL) of sample and standard were prepared by dissolving in methanol and distributed in test tubes. Then, 3 mL of 0.004% *w*/*v* methanol solution of DPPH was admixed with prepared samples solution (2 mL) of the and left in a dark place for 30 min. The absorbance of different solutions was determined at 517 nm employing a spectrophotometer. The percentage of inhibition activity was calculated using the following equation:%I=(A°−Aϰ)/A°×100
where A° represents the absorbance of the control, Aϰ represents the absorbance of the extract or the standard solution. Calculation of IC_50_ was done by plotting the % inhibition against the concentration.

### 4.8. Antiproliferative Activity

#### 4.8.1. Animals

This study was conducted according to standard ethical guidelines. The Research and Ethical Committee of the Applied Science Private University approved all the experimental protocols (Approval Number: 2015-PHA-05). This study was conducted using 48 healthy female Balb/C mice, ranging between 21–25 g of weight and 6–8 weeks of age. Mice were accommodated in well-ventilated rooms, at RT (25 °C) and 50–60% humidity, as well as alternating cycles of dark and light every 12 h, in order to accomplish all required environmental conditions. They were raised in cages equipped with wooden shavings for bedding, a special water bottle, and food.

#### 4.8.2. Cell Lines and Cell Culture Conditions

Different tumor cell lines were used to evaluate the antiproliferation activity of *A. citrodora* extracts. The tested cell lines were as the following: human breast cancer cells (MCF-7, T47D, MDA-MB-231), mouse mammary carcinoma cells (EMT6/P), and normal cells (Vero). The European Collection of Authenticated Cell Cultures (ECACC) (UK) was the source of all cell lines used in the experiments. Optimal cell culturing conditions were considered for cell growth using complete tissue culture media and appropriate incubation environment; 37 °C, 5% CO_2_, and 95% humidity. To prepare a complete tissue culture medium, media were completed with 10% fetal bovine serum, 1% L-Glutamine, 1% Penicillin-Streptomycin solution, 0.1% Gentamycin sulphate solution, and 0.1% Non-Essential Amino acids [125]. Cells were cultured using suitable tissue culture media for MDA-MB-231 was cultured in complete Dulbecco’s Modified Eagle Medium (DMEM). In contrast, T47D and MCF-7 cell lines were cultured in complete Roswell Park Memorial Institute (RPMI) 1640 Medium, EMT6, and Vero cell lines cultured in Gibco Minimum Essential Media (MEM). The toxicity of extracts in vitro was determined by the normal cell line (Vero).

#### 4.8.3. Cytotoxicity and Antiproliferative Activity Assay

Cell viability was measured using the MTT (3-(4,5-Dimethylthiazol-2-yl)-2,5-diphenyltetrazolium bromide) assay kit. After culturing, trypsinization, and cell counting for the chosen cell line, at a concentration of 15,000 cells/well, cells were dispensed 100 μL/well into each well of 96-well tissue culture plates. Following incubation for 24 h, the media in each well was removed entirely. The adherent cells were treated in triplicates with decreasing concentrations of *A. citrodora* extracts (5–0.015 mg/mL) to obtain a total volume of 200 μL/well. After a 48-h incubation for plates under specific conditions, media was removed from each well, washed with PBS, and replaced with fresh media, followed by 20 µL of Thiazolyl blue tetrazolium bromide solution and incubated for 3 h. After that, 100 µL of DMSO was augmented for each well to stop the reaction and dissolve the formazan crystals particles formed in viable cells. After incubation for 1 h, the plate was placed on a microplate reader (ELISA) microplate absorbance reader. The optical density was determined at 550 nm. Microsoft Excel 2016 software program (Microsoft Inc., Washington, DC, USA) was utilized to apply further calculations that determine the proportion of survival cells and calculate the IC_50_ values [125].
$$\mathrm{Percentage}\;\mathrm{of}\;\mathrm{Cell}\;\mathrm{Viability}\;(\%)=\frac{\mathrm{OD}\;\mathrm{of}\;\mathrm{treated}\;\mathrm{cell}}{\mathrm{OD}\;\mathrm{of}\;\mathrm{control}\;\mathrm{cell}}\times 100$$

The estimation of the selectivity index depends on the IC_50_ ratio of the normal cell line (Vero cells) versus the tumor cells, which demonstrates the selectivity of the sample to the tested cell lines. The samples are classified high selective if their SI value is more than three [25].
$$\mathrm{SI}=\frac{\mathrm{the}\;{\mathrm{IC}}_{50}\;\mathrm{of}\;\mathrm{the}\;\mathrm{normal}\;\mathrm{cell}\;\mathrm{line}\;\left(\mathrm{Vero}\right)}{{\mathrm{IC}}_{50}\;\mathrm{of}\;\mathrm{each}\;\mathrm{extract}}$$

#### 4.8.4. Acute Toxicity Test of *A. citrodora* Ethyl Acetate Extract

A limit test was conducted on a small group of mice to select the dose ranges for actual LD_50_ (median lethal dose). Ethyl acetate extracts were dissolved in PBS and 5% tween 20. Four female mice (6 weeks old, 20–23 g weight) were injected intraperitoneally (IP) with a plant extract dose (5 g/kg) which was obtained from the literature [126,127]. The mice were observed for 24 hr. for any mortality. The next dose was increased by 1.5 times if tolerated or decreased by 0.7 times if lethality occurred on new animals. The maximum non-lethal and minimum lethal doses were used as lower and upper limits to calculate LD_50_ doses [128]. For LD_50_ determination of A. citrodora, four groups (*n* = 6) of mice have injected IP with several concentrations (2 g/kg, 1.9 g/kg, 1.8 g/kg, 1.7 g/kg). The fifth group takes place as a negative control. The mice were monitored for 24 hr. for mortality and general behavior. The concentration that causes 50% mortality was recorded as LD_50_. The actual LD_50_ was determined using the arithmetical method of Karber [128]: where (a) represents the Dose difference and (b) represents the Mean of mortality.
$${\mathrm{LD}}_{50}=\mathrm{the}\;\mathrm{least}\;\mathrm{lethal}\;-\;\sum (a*b)/\mathrm{number}\;\mathrm{of}\;\mathrm{mice}$$

#### 4.8.5. Antitumor Effect of *A. citrodora* in Mice Model Experiment

In vivo experiments were carried out on female Balb/C mice, each bearing EMT6/P tumors. Dose establishment was based on the LD_50_ determined previously. *A. citrodora* dose calculated for in vivo experiments was (0.16 g/kg/day) to be injected IP for ten days. EMT6/P cells were collected using trypsin-EDTA and assessed for viability via blue exclusion methods. A suspension of 1.5 million cells/mL MEM was prepared, and a tumorigenic dose of 150,000 cells, 100 μL, was injected subcutaneously in the abdominal area of female Balb/C mice. Ten days after inoculation, tumor volumes, also known as (tumor sizes) were measured using a digital caliper to define the length and width. Tumor volume calculation is a helpful technique that provides a quantitative assessment of tumor growth and progression. In this context, Tumor volumes and tumor progression were estimated applying the below equations, respectively [129]:$$V=\frac{{(L\times W}^{2})}{2}$$
where: V, L, and W are the volume, length, and width of the tumor, respectively.
$$\%\mathrm{tumor}\;\mathrm{change}=\frac{F-I}{I}\times 100\%$$F, I represent the final and initial tumor volumes, respectively.

Following the measurement of tumor sizes, mice were randomly divided into two groups, each containing nine mice (Table 10). The initial average volumes of tumors of the treated group almost matched the untreated positive control group, with insignificant differences:

### 4.9. Evaluation of Liver and Kidney Function in Treated Mice

Serum levels of liver enzymes AST and ALT were investigated for mice treated with A. citrodora, untreated mice with tumor, and normal mice without tumor. AST and ALT were assessed using the Aspartate Aminotransferase (AST/GOT) kit and the Alanine Aminotransferase (ALT/GPT) kit. The reagent was mixed, meeting the protocols to prepare working reagents (4 mL of reagent A mixed with 1 mL of reagent B). Working reagents were incubated at 37 °C, the optimal reaction temperature. In a quartz cuvette, 50 µL of each sample was mixed with 1 mL of working reagent, incubated for 1 min, and recorded the initial absorbance. Absorbance readings were also recorded after 0, 1, 2, and 3 min. The spectrophotometer was set to read absorbance at 340 nm.

To assess the toxicity levels exerted on the kidney by different treatments and the possibility of developing nephrotoxicity: creatinine serum levels were investigated for the same mentioned groups using Creatinine Kit. Standard (S) (BioSystems, Barcelona, Spain). The working reagent was incubated at 37 °C, the optimal reaction temperature. one volume of each working reagent was mixed. 100 μL of each sample was mixed with 1 mL of working reagent in a plastic cuvette. Absorbance readings were recorded after 30 s and 90 s. The spectrophotometer was set to read absorbance at 510 nm.

### 4.10. Statistical Analysis

Data were averaged and expressed as the mean ± Standard Error of Mean (SEM) of triplicate independent experiments using the SPSS statistical package version (version 22). SPSS one-way analysis of variance (ANOVA), tukey, and one-sample t-test was used to establish the statistical significance among the groups. Differences between groups were considered significant when the *p*-value was less than 0.05 (*p* < 0.05). IC_50_ values, obtained for the different extracts of *A. citrodora* aerial parts in different cell lines, were calculated using non-linear regression in SPSS statistical package (version 22).

## 5. Conclusions

A. citrodora is a plant with promising antioxidant and anti-proliferation effects. Results revealed effective antioxidant ethanol extract and inactivity of the essential oil primarily due to its uncommon chemical composition. The antiproliferative potential of the extracts was screened on different cell lines, and ethyl acetate extract demonstrated superior antiproliferative activity, especially against EMT-6 and MCF-7 breast cancer cell lines. Experiment in-vivo conducted using the same extract on mice bearing breast cancer. Results revealed a promising significant reduction in tumor size and high curing outcome. Both effects result from the occurrence of biologically active compounds predominantly characterized by naringenin, 5,6,4′-Trihydroxy-7,3′-Dimethoxyflavone, hispidulin, and eupatilin. Nevertheless, more research is required to explore the *A. citrodora* components and their molecular mechanisms of action with additional testing to confirm the antioxidant activity. Additionally, in silico studies must be conducted to link the structure of identified compounds and their mechanisms of action.

## Figures and Tables

**Figure 1 plants-11-00785-f001:**
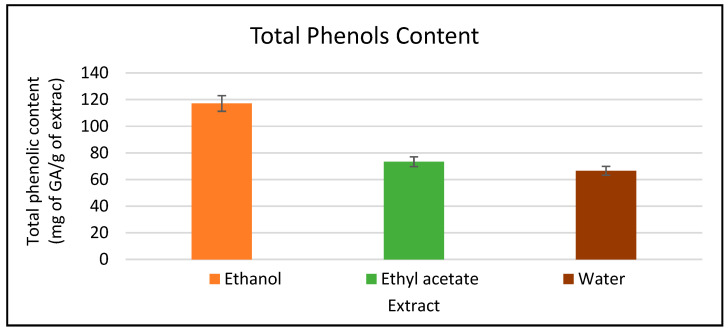
Total phenolic content in mg GAE/g dry weight of *A. citrodora* extracts. Results are expressed as means of three independent experiments (bars) ± SEM (lines).

**Figure 2 plants-11-00785-f002:**
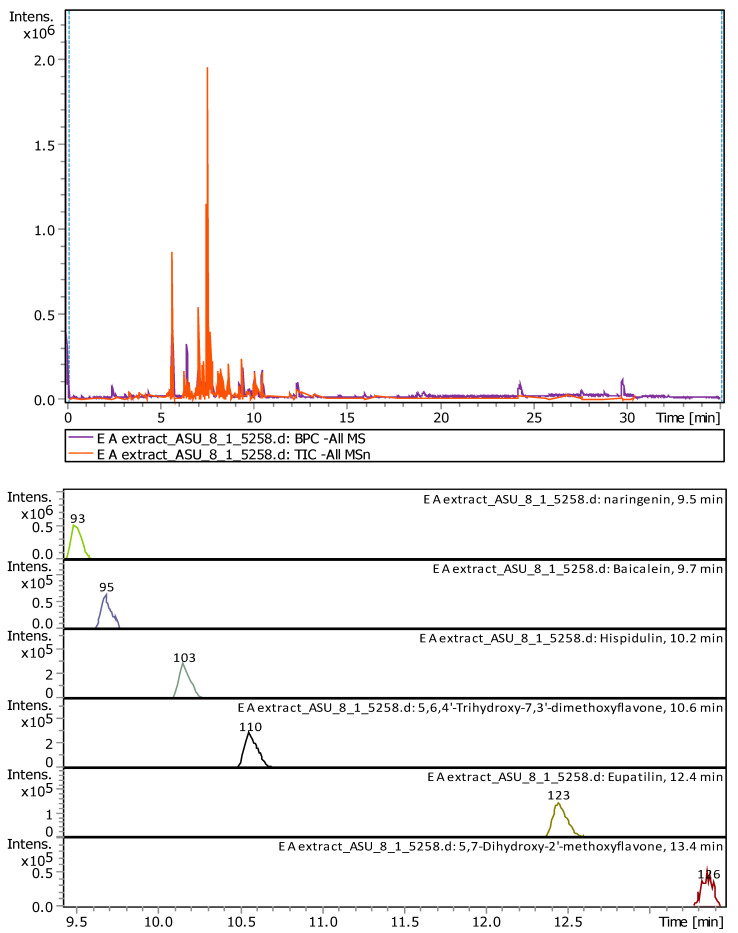
LC-MS Chromatogram obtained from *A. citrodora* ethyl acetate extract. A total of six identified compounds was revealed: Naringenin, 5,6,4′-Trihydroxy-7,3′-Dimethoxyflavone, Hispidulin, Eupatilin, Baicalein, and 5,7-Dihydroxy-2′-Methoxyflavone.

**Figure 3 plants-11-00785-f003:**
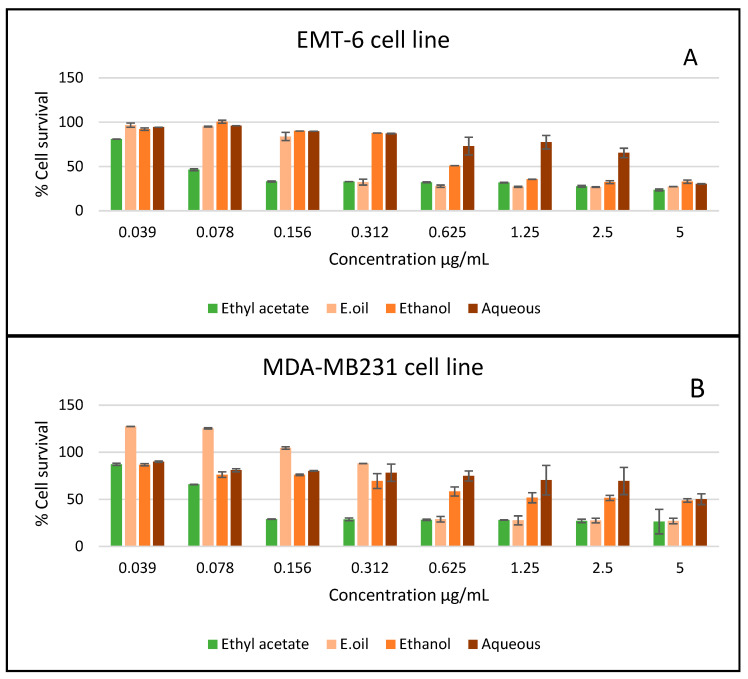
The antiproliferative activity of *A. citrodora* extracts and essential oil against different cancer cell lines: (**A**) EMT6/P, (**B**) MDA-MB231, (**C**) MCF-7, (**D**) T47D. Results are expressed as means of three independent experiments (bars) ± SEM (lines).

**Figure 4 plants-11-00785-f004:**
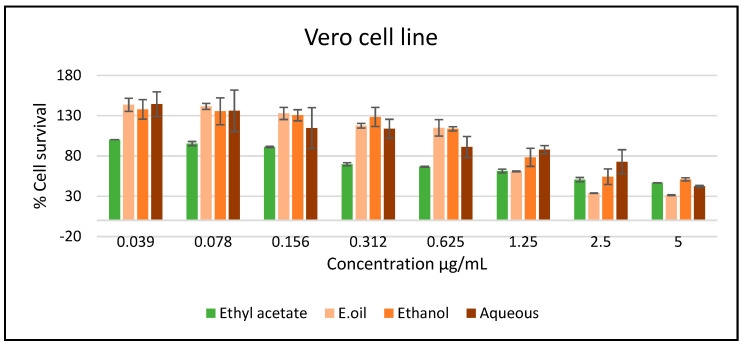
The cytotoxic activity of the *A. citrodora* extracts and essential oil against the Vero cell line. Results represented as mean ± SEM.

**Figure 5 plants-11-00785-f005:**
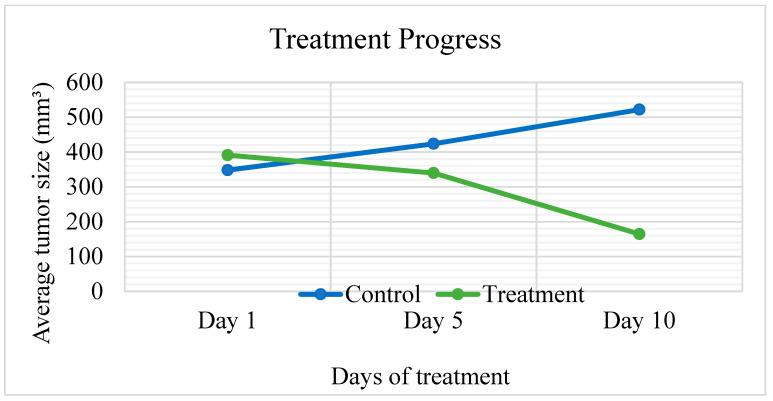
A plot of change in average tumor size (mm^3^) vs. time in (days) of treatment in EMT-6/P cells inoculated in Balb/C mice. (*p* = 0.003).

**Figure 6 plants-11-00785-f006:**
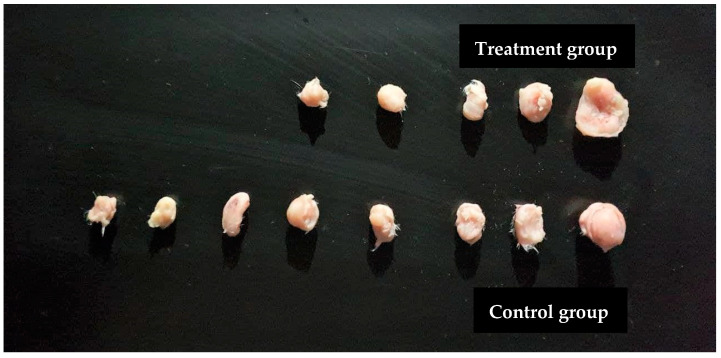
Comparison of tumor sizes after dissection at day 10 in both groups, *n* = 9. Treatment of the mice with ethyl acetate extract of *A. citrodora* resulted in undetected and lower tumor sizes (*p* = 0.003) compared to the control group.

**Table 1 plants-11-00785-t001:** Percentage yields obtained by extraction of 100g of *A. citrodora* using different extraction solvents and methods.

Extraction Solvent	Yield (*w*/*w*%)
Water	5.5%
Ethanol	5.25%
Ethyl acetate	2.3%
Essential oil (HD)	0.25%

**Table 2 plants-11-00785-t002:** Major compounds identified in ethyl acetate extract of *A. citrodora* using LC-MS.

Name	Molecular Formula	Molecular Wt.	Retention Time	% Of the Identified Compounds
Naringenin	C_15_H_12_O_5_	272.06834	9.5	25.22
Baicalein	C_15_H_10_O_5_	270.05286	9.7	7.84
Hispidulin	C_16_H_12_O_6_	300.06345	10.2	22.61
5,6,4′-Trihydroxy-7,3′-Dimethoxyflavone	C_17_H_14_O_7_	330.07387	10.6	23.67
Eupatilin	C_18_H_16_O_7_	344.08932	12.4	13.24
5,7-Dihydroxy-2′-Methoxyflavone	C_16_H_12_O_5_	284.06833	13.4	4.91

**Table 3 plants-11-00785-t003:** Components of the volatile oils of the fresh aerial parts of *A. citrodora* obtained by HD and SPME.

RI ^1^	Compound	HD ^2^ %	SPME ^3^ %
938	α-Pinene	0.85	-
975	Sabinene	-	4.00
974	*β*-Pinene	2.10	0.44
980	Octanone (3-)	0.34	-
986	Cineol <dehydro-1,8->	0.55	-
991	Myrecene	0.20	-
1032	d,l-Limonene	18.80	34.40
1034	Delta-3-Carene	0.13	-
1035	Cineol <1,8->	8.14	-
1046	Ocimene <(E)-beta->	1.19	5.04
1070	Sabinene hydrate <cis->	0.20	-
1073	Mentha-3,8-diene <para->	0.39	-
1099	Linalool	0.38	-
1130	Allo-ocimene	-	1.72
1164	*cis*-Chrysanthenol	0.50	-
1183	*cis*-Pinocarveol	0.70	-
1185	α-Terpineol	0.65	-
1196	Myrtenol	0.21	-
1200	γ-Terpineol	1.98	-
1229	Dihydrocarveol <neoiso->	1.90	-
1240	*trans*-Chrysanthenyl acetate	10.27	-
1255	*cis*-Myrtanol	1.82	-
1277	Verbenyl acetate	9.10	-
1296	Geranyl formate	-	0.64
1334	δ-Elemene	-	1.74
1377	α-Copaene	1.06	4.17
1380	Geranyl acetate	0.42	-
1385	*β*-Cubebene	0.63	1.23
1403	α-Longipinene	-	0.27
1419	α-Santalene	0.89	1.25
1422	*β*-Caryophyllene	5.09	16.80
1427	*β*-Copaene	0.71	1.18
1441	α-Guaiene	-	1.35
1444	Aromadendrene	-	0.80
1448	Muurola-3,5-diene <cis->	-	0.12
1451	α-Himachalene	0.15	-
1458	α-Humulene	0.38	0.92
1460	α-Patchoulene	0.95	0.17
1462	Allo-aromadendrene	-	2.20
1471	*β*-Acoradiene	0.44	0.92
1477	γ-Gurjunene	0.30	1.76
1483	γ-Muurolene	14.13	17.25
1484	γ-Curcumene	-	0.21
1496	γ-Amorphene	1.00	-
1495	Viridiflorene	-	0.23
1498	Bicyclogermacrene	2.39	
1511	α-Cupranene	-	0.58
1515	γ-Cadinene	0.36	0.20
1517	*β*-Curcumene	0.20	
1520	*δ*-Cadinene	0.82	0.22
1563	Germacrene B	0.34	-
1582	Spathulenol	3.75	-
1585	Caryophyllene oxide	2.65	-
1595	Viridiflorol	0.20	-
1647	α-Muurolol	0.60	-
	Total identified	97.86	99.81
	Terpenoids	97.52	99.81
	Monoterpenes	60.48	46.24
	Monoterpene hydrocarbons	23.66	45.6
	Oxygenated monoterpenes	36.82	0.64
	Sesquiterpenes	37.04	53.57
	Sesquiterpene hydrocarbons	29.84	53.57
	Oxygenated sesquiterpenes	7.2	-
	Non-terpenoids	0.34	-

^1^ RI: retention index, ^2^ HD: hydrodistillation, ^3^ SPME: solid phase microextraction.

**Table 4 plants-11-00785-t004:** IC_50_ of different extracts of *A. citrodora* against DPPH.

Extract	IC_50_ mg/mL
Water	49.918
Ethanol	22.858
Ethyl acetate	107.044
Hydrodistilled oil	-

**Table 5 plants-11-00785-t005:** IC_50_ (µg/mL) for different extracts of *A. citrodora*.

	IC_50_ (µg/mL) ± SEM				
	EMT-6	MCF-7	MDA-MB231	T47D	Vero
Ethyl acetate	138 ± 20	136 ± 13	203 ± 60	180 ± 40	840 ± 160
Essential oil	410 ± 20	540 ± 40	633 ± 14	402 ± 10	1990 ± 100 ^1^
Ethanol	730 ± 300	1510 ± 420 ^1^	1240 ± 291 ^1^	1076 ± 460 ^1^	2510 ± 320 ^1^
Aqueous	4330 ± 460 ^1^	2180 ± 380 ^1^	5000 ± 70 ^1^	3780 ± 730 ^1^	4540 ± 1540 ^1^

^1^ IC_50_ > 1000 µg/mL is considered to be inactive.

**Table 6 plants-11-00785-t006:** The selectivity index for *A. citrodora* extracts and essential oil. SI = IC_50_ of the normal cell line (Vero)/IC_50_ of each extract.

	SI ^1^ for the Active *A. citrodora* Extracts and Essential Oil		
	Ethyl Acetate	Essential Oil	Ethanol
EMT-6	6.08	4.85	3.85
MCF-7	6.17	3.68	- ^2^
MDA-MB231	4.13	3.14	- ^2^
T47D	4.66	4.95	- ^2^

^1^ SI referred to Selectivity Index when SI value >3 indicates high selectivity. ^2^ inactive.

**Table 7 plants-11-00785-t007:** Results for LD_50_ using the Karber method.

Group	Dose g/mL	No. of Mice	No. of Dead Mice	Dose Difference (a)	Mean of Mortality (b)	a∗b
vehicle	control	6	0	0	0	0
1	1.7	6	2	0.1	0.5	0.05
2	1.8	6	2	0.1	1	0.1
3	1.9	6	3	0.1	1.5	0.15
4	2	6	3	0.1	2	0.2
SUM						0.5

**Table 8 plants-11-00785-t008:** *A. citrodora* effect on tumor size, weight, cure, and deaths percentages, in EMT-6/P cell line. Where (mm^3^) is a cubic millimeter (*n* = 9). ^1^ (*p* = 0.656), ^2^ (*p* = 0.003).

	Av. Initial Tumor Size (mm^3^) ± SEM	Av. Final Tumor Size (mm^3^) ± SEM	% Change in Tumor Size	% Of Mice with No Detectable Tumor	Average Tumor Weight (g)
Control	348.12 ± 1.23	521.98 ± 0.43	49.94	11.11	443.22
Treatment	391.14 ± 1.12 ^1^	164.38 ± 0.23	−57.97 ^2^	44.44	285.33

**Table 9 plants-11-00785-t009:** Serum ALT, AST levels (IU/L), and Serum creatinine levels (µmol/L) for treatments, control, and normal untreated mice groups.

Group	ALT (IU/L) ± SEM	AST (IU/L) ± SEM	Creatinine (µmol/L) ± SEM
Normal mice	28.88 ± 9.42	119.98 ± 9.42	97.24 ± 1.76
Control	37.21 ± 13.88	101.65 ± 10.55	106.20 ± 37.88
Treatment	21.66 ± 7.22 ^1^	98.32 ± 27.21 ^2^	207.23 ± 113.65 ^3^

^1^ (*p* > 0.453), ^2^ (*p* > 0.92), ^3^ (*p* > 0.532).

**Table 10 plants-11-00785-t010:** In vivo experimental groups and the treatments received.

GROUP	DOSE
Control	Untreated
*A. citrodora* Ethyl acetate extract	(0.16 g/kg/day) IP for 10 days

*n* = 9 for each group.

## Data Availability

Data is contained within the article or Appendix A.

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
