# Peer review of "Antioxidant and Antiproliferation Activities of Lemon Verbena (*Aloysia citrodora*): An In Vitro and In Vivo Study"

_plants, 2022, doi:10.3390/plants11060785_

Round 1

Reviewer 1 Report

Plants (Manuscript ID: plants-1610426), Comments to the Authors:

Title: Antioxidant and Anticancer Activities of Lemon verbena (Aloysia citrodora): An In-Vitro and In-Vivo Study 

Comments:

The submitted paper discussed the antioxidant and cytotoxic effects of Lemon verbena (Aloysia citrodora). In addition to volatiles extraction, different solvent extracts were prepared. GC-MS, LC-MS analysis, and Foline–Ciocalteu (F-C) assays were used to identify the plants’ chemical composition. MTT assay was used to measure the antiproliferative ability for each extract. Antioxidant activity was determined using the 2,2-diphenylpicrylhydrazyl (DPPH) assay. In in vivo cytotoxic experiments, Balb/C mice were inoculated with tumor cells and treated with ethyl acetate extract of A. citrodora. After treatment, a significant reduction in tumor size (57.97%) and high cure percentages (44.44%) were obtained in treated mice, demonstrating the antiproliferative efficacy of the ethyl acetate extract. On the other hand, ethanol extract showed the most potent antioxidant activity.

I think the submitted manuscript can be accepted for publication after the authors respond to the following comments:

  1. several manuscripts analyzed the cytotoxic and antioxidant activities of Aloysia citrodora. The authors should compare their results with previous reports and show the novelty of their work.
  2. The authors mentioned the anti-cancer effect. This term should be used only for drugs tested in clinical trials.
  3. What do the authors mean by high cure percentage in the abstract? The term is not clear.
  4. The authors did not show the rationale behind the selection of their plant. Why did they select this plant What is the knowledge gap the authors tried to fill by doing their research.
  5. Figure 5 is not clear. The authors should indicate what are the tumor samples in the figure.
  6. The authors should include chromatograms in their manuscript.

Author Response

Thank you for your comments and positive feedback. Attached is the detailed response.

We hope that this modified manuscript will meet your expectations.

Reviewer 2 Report

The authors present an experimental study investigating the antioxidant and anticarcinogenic activity of extracts from the plant lemon verbena (Aloysia citrodora). The authors tested water, ethanol, and ethyl acetate extracts, as well as the essential oil obtained by hydrodistillation. GC-MS, LC-MS and FC were used to identify the compounds found in the extracts, while the DPPH assay was used to determine antioxidant activity. They also performed antiproliferative measurements using the MTT assay and in vivo experiments for anticarcinogenic properties in Balb/C mice. They showed good anticarcinogenic activity of ethyl acetate extract and good antioxidant activity of ethanolic extract. The study seems well designed and is important because of the increasing importance of natural products in drug development. However, I have comments that would require a major revision of the work and I list them below.

  • The overall English and sentence formation need improvement, some sentences are not easy to understand, such as the one on page 9, line 254: "Ethyl acetate extract showed ... ".
  • What I miss most in the paper is a thorough discussion of the mechanisms behind the activity of the known compounds against cancer cells (mechanisms known from literature). It would be good if you could relate the most representative compounds from lemon verbena to their known anticancer pharmacological activity, as this is currently done only briefly in your work (a good and clear presentation of the properties of many polyphenolic compounds is described in Molecules 2016, 21(7), 901). Baicalein, for example, which you mention, is known to cause cell cycle arrest in the G2/M phase. Moreover, a discussion of the anticarcinogenic properties of specific flavonoids can be found at Foods 2021, 10(6), 1331. In addition, there is an abundance of recently published in silico studies linking the activity of natural products to their potential mechanisms, which is not addressed in your study (Foods 2022, 11(1), 67). Similar studies should be available for representative compounds in lemon verbena, and in silico studies in general have great potential in identifying the mechanisms behind health promoting effects of natural based compounds.
  • As in the study mentioned, can you address possible metabolic changes in the representative compounds (if known). Overall, pharmacokinetic properties can have a major impact on the health-promoting effects of phytochemicals. You have already briefly addressed this issue by discussing the presence or absence of sugar moieties.
  • There is also currently only a very brief discussion of the antioxidant mechanisms of the lemon verbena compounds. It would be good if the authors could use an example mechanism to show in a scheme how a representative compound can scavenge free radicals. Again, there are already in silico studies with quantum mechanical calculations on how plant compounds can react with free radicals Antioxidants 2020, 9(7), 587. Please discuss how the presence of functional groups (OH) of the compounds affects the radical scavenging function. Based on this, you can determine which compounds are most likely to have good antioxidant activity. Please show the structures of the most representative compounds and which parts are most likely to react with radicals. For a discussion of how another plant compound carnosic acid (from rosemary) may lead to antioxidant mechanisms, see again Foods 2022, 11(1), 67 (Introduction).
  • Why was LC-MS performed only on the ethyl acetate extract?
  • Whereas FC is a good method for measuring the total phenolic content, there are some important downsides to it. Please discuss them. For example, again see Foods 2022, 11(1), 67.
  • Similarly, DPPH is a widely used method for antioxidative potential determination, but how can it compare to some other methods, such as ABTS.
  • Please compare in more detail the composition of your extracts/essential oils to others reported in the literature for lemon verbena.
  • Please connect the polarity of representative compounds to their presence in specific extracts.

Author Response

Thank you for your comments and positive feedback. Attached is the detailed response.

We hope that the modified manuscript will meet your expectations

Reviewer 3 Report

In the present manuscript, the antioxidant and anticancer effects, as well as the chemical composition of Aloysia citrodora were evaluated.

Table 1: remove the second column since it is the same as column 3.

Table 3: present all the detected compounds.

What is the difference in the results presented in Table 5 and Figure 2. IC50 represents the concentration that causes 50% inhibition. Therefore, the results should be in accordance. However, the concentration for the 50% survival rate seem to be lower than the IC50 values. Moreover, in Figure 2B,C,D there are some bars that exceed 100%. What does this actually mean? Did the number of cells increase? The same applies for Figure 3.

Lines 209-211. The authors did not examine the composition of ethanol extracts but those of ethyl acetate. They only performed the analysis of TPC. So, the results cannot be justified by this argument.

The authors should explain better why they did not detect citral or any other major compounds that are usually detected in the essential oil of the species (e.g. thujone, citronellal, carvone etc.). Maybe the preparation of the samples prior to extraction had an effect on the obtained composition. 

The evaluation of antioxidant activity with only one assay is not adequate.

The authors should perform a detailed analysis of the chemical composition of all the tested extracts. 

Author Response

Thank you for your comments and positive feedback. Attached is the detailed response.

We hope that the revised manuscript will meet your expectations

Reviewer 4 Report

This study is the result of studying the antioxidant and anticancer efficacy of Different extracts of A. citrodora. In particular, we conducted in vitro and in vivo studies for anticancer research.

The experimental designs are not fine. Authors must provide all experimental protocols in detail with relevant literature.

The extract preparation from A. citrodora, seeded cell densities, sources of cell lines (where authors have obtained these cell lines), used media name, and other relevant details are required in the method section.

Perform and provide statistically significant values for all experimental results. Details legends are required for all figures.

Oxidative stress results from an imbalance between the production of reactive oxygen species (ROS) and their removal by the cellular antioxidant system. I recommend putting it in the results section for intracellular ROS levels of cancer cell lines.

Indeed, the role of A. citrodora extracts in anti-oxidant and cancer has been previously described in detail ( Antioxidant and Anti-Proliferative Activity of Essential Oil and Main Components from Leaves of Aloysia polystachya Harvested in Central Chile. Molecules. 2020 Dec 30;26(1):131. doi: 10.3390/molecules26010131. Chemometric Profiling and Bioactivity of Verbena (Aloysia citrodora) Methanolic Extract from Four Localities in Tunisia. Foods. 2021 Nov 24;10(12):2912. doi: 10.3390/foods10122912. .), the results are lack of novelty.  I recommend putting it in the discussion section about differences from this paper.

Interpret the current data with previously published literature. The present results and discussion are not sufficient.

Author Response

Thank you for your comments and positive feedback. Attached is the detailed response.

We hope that this modified manuscript will meet your expectations

Round 2

Reviewer 1 Report

After reading the authors' response to my comments, I think the revised manuscript can be accepted for publication. 

Author Response

Thank you very much for your positive feedback and valuable comments that increased the quality of our work.

Reviewer 2 Report

The authors successfully addressed all issues raised by this reviewer. Consequently, the manuscript has been significantly improved and can be in its current version recommended for publication in Plants.

Author Response

Thank you very much for your positive feedback and valuable comments that improved the quality of our work

Reviewer 3 Report

The authors addressed most of my comments. However there is still some claification needed in the following issues:

Comment: Moreover, in Figure 2B,C,D there are some bars that exceed 100%. What does this actually mean? Did the number of cells increase? The same applies for Figure 3.

Authors' response: Some bars may give results more than 100% because some plant extracts contains phytochemicals with mitogenic activity. When such phytochemical presents in high concentrations, the proliferation may increase resulting in percentage survival more than 100%.

New comment: Can you give some references for this? If the tested extracts have mitagenic effects, then the increasing concentration should have more severe effects and the survival rate should further increase.

Comment: The authors should explain better why they did not detect citral or any other major compounds that are usually detected in the essential oil of the species (e.g. thujone, citronellal, carvone etc.). Maybe the preparation of the samples prior to extraction had an effect on the obtained composition.  

Authors' response: Thank you for this comment. Various factors may influence the composition of the extract including age stage, growth condition, and physiology. The manuscript was modified to include justification for the phytochemical composition of the plant.

New comment: The authors should provide more details about the growth stage of the tested plants. Provide also a chromatograph of GC-MS. Did the authors used only the calculated relative peak areas for the quantification of compounds?

Lines 416-427: the methodology should be more clear because in Line 418 the authors mention that EO was obtained by hydrodistillation, then (Lines 418-422) they describe the solvent extraction and then (Lines 423-427) the describe the hydrodistillation with Clevenger apparatus.

Author Response

Thank you very much for your valuable comments that enhanced the quality of our work. All your comments and suggestions were considered in the revised manuscript and a detailed response was attached.

We hope that our revised manuscript will meet your expectations.

Reviewer 4 Report

This revised manuscript has been improved as per the instruction of reviewer. I think this manuscript is acceptable for publication.  

Author Response

Thank you very much for your positive feedback and valuable comments that improved the quality of our work.

Round 3

Reviewer 3 Report

The authors addressed all my comments. Therefore, I recommend the acceptance of the manuscript in its present form.